# A Theoretical and Empirical Investigation of Design Characteristics in a Pb(Zr,Ti)O_3_-Based Piezoelectric Accelerometer

**DOI:** 10.3390/s20123545

**Published:** 2020-06-23

**Authors:** Min-Ku Lee, Seung-Ho Han, Jin-Ju Park, Gyoung-Ja Lee

**Affiliations:** Sensor System Research Team, Korea Atomic Energy Research Institute, Daejeon 34057, Korea; leeminku@kaeri.re.kr (M.-K.L.); shh4617@kaeri.re.kr (S.-H.H.); jinjupark@kaeri.re.kr (J.-J.P.)

**Keywords:** piezoelectric accelerometer, compression-mode, finite-element method, Pb(Zr,Ti)O_3_, piezoelectric analysis, metamodel

## Abstract

A theoretical and experimental study on the design-to-performance characteristics of a compression-mode Pb(Zr,Ti)O_3_-based piezoelectric accelerometer is presented. Using the metamodeling to approximate the relationship between the design variables and the performances, the constituent components were optimized so that the generated electric voltage, representing sensitivity, could be maximized at different set values of the resonant frequency (25–40 kHz). Four kinds of optimized designs were created and fabricated into the accelerometer modules for empirical validation. The accelerometer modules fabricated according to the optimized designs were highly reliable with a broad range of resonant frequency as well as sufficiently high values of charge sensitivity. The fixed (or mounted) resonant frequency was between 16.1–30.1 kHz based on the impedance measurement. The charge sensitivity decreased from 296.8 to 79.4 pC/*g* with an increase of the resonant frequency, showing an inverse relation with respect to the resonant frequency. The design-dependent behaviors of the sensitivity and resonant frequency were almost identical in both numerical analysis and experimental investigation. This work shows that the piezoelectric accelerometer can be selectively prepared with best outcomes according to the requirements for the sensitivity and resonant frequency, fundamentally associated with trade-off relation.

## 1. Introduction

Piezoelectric acceleration sensors (or accelerometers) built using Pb(Zr,Ti)O_3_ (PZT) ceramics are widely used in the field of structural health monitoring owing to outstanding characteristics such as a fast response, excellent dynamic range and linearity, long term stability and minimal power consumption [1,2,3]. In detecting abnormal vibrations of structural components, such accelerometers perform best with a high sensitivity (low threshold) and a high resonant frequency (wide frequency usable range) simultaneously. Unfortunately, all piezoelectric type accelerometers are basically a compromise between sensitivity and resonant frequency. To be specific, these two performance characteristics are fundamentally associated by a trade-off relation: namely, one property must increase at the sacrifice of the other [4,5,6,7,8,9]. In considering the possible presence of alternative designs of the sensor components, therefore, challenges always exist in seeking advanced sensor designs with improved performances related to these two characteristics.

Our recent numerical investigation found that in constructing an accelerometer, the performances of these two characteristics were significantly influenced by the design factors (geometry, dimensions and configuration) of the components (e.g., seismic mass, piezo element, tail, base and insulating layer etc.) [10]. To obtain better performances, moreover, it was necessary to carefully choose the component design by optimizing the design variables of each component, which was possible by thoroughly understanding the relationship between the design characteristic and the physical functionality. The numerically analyzed results also revealed that the structure of the accelerometer could be designed according to the requirements of the sensing application.

In this study, we extended our previous numerical work [10] to a real system, offering important information about the role of the design characteristics on the physical functionalities of a compression-mode accelerometer. Given the valid range of the meta model produced using the design of experiments (DOE) technique [10], we chose four set values of resonant frequency between 25 and 40 kHz to numerically optimize the accelerometer design via metamodeling. We obtained the four sets of accelerometer designs with a variation of the resonant frequency and successfully fabricated and assembled the accelerometer modules using the proposed designs. We investigated both theoretically and experimentally their performance characteristics related to the two performance indicators, sensitivity, and frequency response.

## 2. Materials and Methods

As shown in Figure 1, the basic structure of a compression-type piezoelectric accelerometer was employed as a design reference for the numerical analysis. Two piezoceramic rings are connected electrically in parallel and mechanically in series. They are sandwiched under a compressive force between the head and tail by a screw.

The numerical analysis was performed via harmonic and piezoelectric analysis using a commercial finite element modeling software (ANSYS V18). The modeling was performed for a piezoelectric accelerometer module under two boundary conditions, free and fixed conditions. We selected ten design variables of accelerometer module for numerical analysis: head outer diameter (OD), head height, piezoceramic OD, piezoceramic inner diameter (ID), piezoceramic thickness, tail OD, tail height, insulating layer thickness, base OD and base height. The materials used for the head, piezoceramic ring, tail, base, and insulating layer were tungsten, PZT, 316 stainless steel and epoxy, respectively. The kriging model was adopted for building the approximation of the relationship between the design variables (input) and the performances (output) and used for optimization of design variables. Its mathematical description is given elsewhere [11,12,13]. The optimization was carried out to find the optimum values of the employed design variables using a program called Easy Design, in a way that maximize the electric voltage, representing sensitivity, at different set values of resonant frequency. Here, the electric voltage was obtained assuming 1 *g* (=9.8 m s^−2^) of constant gravitational acceleration at 159 Hz (angular frequency *ω* = 1/2π*f* = 0.001). The optimization process was iterated until a satisfactory result was achieved. Details on these optimization processes as well as all the material data used for numerical analysis were described in ref. [10].

For the experimental investigation of the numerically analyzed results, accelerometer sensor modules were fabricated by assembling internal sensor components and piezoceramic rings manufactured with the same materials and designs as those proposed from the numerical optimization. The frequency response of the accelerometer modules was characterized by measuring the electrical impedance and phase versus frequency spectra using a precision impedance analyzer (HP 4294A; Agilent; Santa Clara, CA, USA) to find whether there is any distinct resonance in the frequency range of 0.04–50 kHz. The applied test signal was 0.5 V (DC bias = 0 V). The measurements were carried out under both the free and fixed conditions to compare with the numerically analyzed results.

The frequency response characteristic was also measured via vibration tests under fixed (or mounted) condition of accelerometer using a vibration exciter (SE-9; SPEKTRA, Dresden, Germany), coupled with a function generator and data acquisition system (m + p VibPilot; m + p International, Hannover, Germany), a signal conditioner (Model 133; Endevco, San Juan Capistrano, CA, USA) and a power amplifier (PA 14-500; SPEKTRA). The sensor calibration was made by an internal high-frequency reference accelerometer (352A60; PCB, New York, NY, USA), which is shear-type with a sensitivity of 10 mV/g and a resonant frequency of 95 kHz. The measurements were conducted in the frequency range of 0.01–40 kHz with an applied acceleration of 1 *g*. The vibration systems were controlled by the Microsoft Windows-based software (m + p VibControl; m + p International).

The charge sensitivity was measured using a portable accelerometer calibrator (28959FV; Endevco) that included a built-in vibration exciter, signal generator, computer-controlled amplifier/servo mechanism and so on. The applied range of acceleration was between 1–10 *g*. The sensitivity was measured at a constant frequency *f* of 159 Hz.

## 3. Results

In this study, we optimized the design variables of the PZT-based accelerometer using the DOE test points used in our previous investigation [10]. According to the results, the present metamodel was valid between 25–47.5 kHz in resonant frequency due to the dimensional limits of size range of the design variables. The optimization was thus carried out by employing four different set values of resonant frequency *f*_s_, 25, 30, 35, 40 kHz, thereby creating four accelerometer designs according to the resonant frequency. To avoid the complexity involved in modeling different geometric dimensions of piezoceramic samples as well as the difficulty in experimentally preparing piezoceramic samples of different sizes or shapes with identical physical properties, the dimensions of piezoceramic ring were kept constant: 12.6 mm for the O.D., 7.5 mm for the I.D. and 2.7 mm for the thickness. All of the other design variables except for those related to the piezoceramic ring were considered for the optimization. The optimized results of the design variables at each *f*_s_ as well as the corresponding results of resonant frequency *f*_r_ and electric voltage *E*_v_ are presented in Table 1 and Figure 2. The results optimized at each *f*_s_ clearly indicate that the *E*_v_ changes in inverse proportion to the *f*_r_ with exponential decay. As highlighted by underlining in Table 1, four results showing the *f*_r_ value closest to *f*_s_ were taken among them for further optimization.

Since the epoxy layer was known to be sensitive to the sensing performances [10], next we investigated the isolated effect of the epoxy used as the insulating layer on the resonant frequency and electric voltage. For this, we considered the size range of the epoxy thickness, 0.3–1.0 mm: About 0.3 mm is considered as minimum thickness that can be experimentally controlled for insulating. Using the above underlined results (Table 1), only the epoxy thickness was varied in this size range. An increase in the insulating epoxy thickness negatively affected the resonant frequency (Figure 3a), while its effects on the electric voltage was negligible (Figure 3b). Thus, the results revealed that a smaller thickness is preferable for enhancing the sensing performance. We also note the results of Figure 3c showing the effect of the epoxy property, i.e., Young’s modulus. As the Young’s modulus of the epoxy increases, the value of *f*_r_ increases until it approaches 5–10 GPa but the increase of *f*_r_ is insignificant beyond this range. According to the result, we confirmed that the epoxy used in this study possesses a desirable property of Young’s modulus (~9.5 GPa) in terms of resonant frequency. Finally, we obtained the optimum thickness of the epoxy layer, i.e., 0.3 mm. Figure 4a presents the *f*_r_ and *E*_v_ properties of the final four designs obtained before and after the numerical optimization of the epoxy layer. It is found that the values of *f*_r_ were increased by about 1.2–1.7 kHz by optimizing the epoxy layer, as compared to those underlined in Table 1. The cross-sectional views of corresponding final designs and their dimensional properties are presented in Figure 4b and Table 2, respectively.

Figure 5 shows the impedance-frequency characteristics obtained from the numerical simulation under free and fixed boundary conditions for the four optimized designs. As determined from Figure 5, the values of *f*_r_ for the fixed mode decreased by about 9.4–10.6 kHz, compared to the free mode (Table 2). This indicates that the *f*_r_ value decreases when the accelerometer is attached or mounted on certain structures for practical application.

For experimental validation of the above numerical results, the accelerometer modules were fabricated using the components and PZT ceramic rings manufactured in accordance with the four optimized designs (Table 2). Including the head, tail, base, and PZT ceramic rings, the fabricated internal components for the four different optimized designs are shown in Figure 6a. The four accelerometer modules assembled using them are also shown in Figure 6b. The epoxy thickness was constantly maintained in the range of 0.3–0.34 mm by precisely controlling the amount of injected epoxy using a specially designed injection mold. After the alignment of the manufactured sensor components, all the components were tightened with a screw using an optimized torque value of about 2.0 N⋅m [10]. As summarized in Table 3, the PZT ceramic rings used in this study had a static piezoelectric coefficient *d*_33_ of 400 pC/N and a Curie temperature *T*_c_ of 367 °C, which are comparable to those of the commercial PZT 5A that are widely used for piezoelectric-typed sensors [14]. The dimensions of the PZT ceramic rings were also nearly the same as those used for the above numerical analysis.

The frequency response was first evaluated through an impedance test and a vibration experiment. To compare with the numerically analyzed results, both the free and fixed conditions were realized in a real experiment. Figure 7a,b show the impedance spectra measured under the free and fixed conditions and the frequency response characteristics obtained from the vibration experiments under the fixed condition, respectively. The *f*_r_ values determined from those experiments are presented in Figure 7c as a comparison with the numerical ones. According to the impedance test, the experimentally measured *f*_r_ values for the fixed condition were lower by 7.3–10.3 kHz than those for the free condition, which was similar to the above numerical investigation (Figure 5).

On the other hand, it is noted that the experimental values of *f*_r_ were slightly lower than the numerical ones. The differences were about 2.3–4.5 kHz for the free condition and about 0.9–3.2 kHz for the fixed condition. One of the reasons for such differences might be related to the difficulties in uniformly maintaining the epoxy thickness at the numerical size of 0.32 mm. We also speculated that it may be related to loose boundaries or interfaces inside the real system, which was different from the numerical analysis assuming the perfect contacts between the components. There being no significant difference in the fixed *f*_r_ values between the impedance test (16.1–30 kHz) and the vibration experiment (14.7–29.3 kHz) proves the reliability of the proposed accelerometer modules. Importantly, these values of *f*_r_ under the fixed condition are substantially high compared to those of commercial piezoelectric accelerometers (20–30 kHz).

We investigated the sensitivity properties of the four optimized accelerometer modules using a portable accelerometer calibrator. Figure 8a shows the charge-to-acceleration characteristics depending on the accelerometer design. In the investigated acceleration range (1–10 *g*), the values of Pearson’s correlation coefficient *r*, as a measure of the linear correlation between two variables, were found to be almost “unity” with perfect linearity, indicating an excellent reliability of the proposed accelerometer modules. The charge sensitivity *S*_q_ can be determined from the slope or the linear relation between them. The obtained *S*_q_ varied from 296.8 to 79.4 pC/*g*, depending on the design (*f*_s_ value). The design dependency of the experimentally measured *S*_q_ is consistent with that of the numerically calculated *E*_v_ (Figure 8b). By considering only the piezoceramic and mass (head), the charge sensitivity was simplified and given by Equation (1) [15].
*S*_q_ = *n* ⋅ *d*_33_ ⋅ *m*_s_ ⋅ *g*(1)

Here, *n*, *m*_s_, and *g* are the number of piezoceramic layers, the weight of the seismic mass, and the acceleration, respectively. Hence, the *S*_q_ is proportional to the design parameters *n* and *m*_s_ as well as the material characteristic *d*_33_. In this work, the *m*_s_ (weight of the head) varied depending on the design, because the dimensions of the head (head OD *x*_1_, head height *x*_2_) changed with the design (Table 2). Given that other parameters, such as *n* and *d*_33_, are fixed, the comparison between the experimentally measured *S*_q_ and *m*_s_ showed that there is a direct proportional relationship between them (Table 4), which is in good agreement with Equation (1). On the other hand, *m*_s_ negatively affects the resonant frequency *f*_r_. Therefore, a change in *m*_s_ must produce the opposite effect on the sensitivity and resonant frequency. The *f*_r_-dependent behavior of *S*_q_ calculated by Equation (1) was also identical to that of the experimentally measured one. However, we also note that there is a difference in their absolute values: the measured values were lower by 73–79% compared to the calculated ones. This might have been due to the presence of other components (e.g., tail, base, epoxy, etc.) and interfaces negatively affecting the sensitivity property in a real sensor system. The results of the fixed *f*_r_ and the corresponding *S*_q_ (or *E*_v_) values clearly showed that there is an inverse relationship between the sensitivity and resonant frequency. Consequently, the design variables can only improve one of the two performances, i.e., sensitivity or resonant frequency, but with sacrifice to the other, when the material properties are fixed. According to our results, we also recommend that the accelerometer design be constructed according to the desired performance on the line of the optimized curve, *S*_q_ versus *f*_r_. For example, either a design with high sensitivity as well as relatively low resonant frequency can be chosen by requirement, or a design with high resonant frequency at the expense of sensitivity may be favored for a wider spectrum of vibration sensing.

## 4. Conclusions

From design to performance, we fully investigated a compression-mode Pb(Zr,Ti)O_3_-based piezoelectric accelerometer both theoretically and experimentally. The free and fixed modes of operation were considered for both numerical and experimental studies. Through the numerical optimization of design variables of the accelerometer with respect to the sensitivity and resonant frequency properties, four kinds of accelerometer designs were created. The design-to-performance characteristics obtained by the numerical analysis were demonstrated by fabricating the accelerometer modules with the proposed designs. The accelerometer modules exhibited a broad range of fixed (or mounted) resonant frequency between 16.1–30.1 kHz based on the impedance measurement. The corresponding values of charge sensitivity varied from 296.8 to 79.4 pC/*g*. The correlation indicated that the two properties had an inverse proportion relation. Since these two main performances are inextricably connected by a trade-off relation, our numerical and empirical study shows that both performances need to be maximized through design optimization of the constituent components that have positive or negative effects, including opposite effect (e.g., mass), on both performances. In addition, this study also suggests that the piezoelectric accelerometer can be selectively prepared using the optimized designs in accordance with the usage objective or the requirements with regards to the sensitivity and resonant frequency.

## Figures and Tables

**Figure 1 sensors-20-03545-f001:**
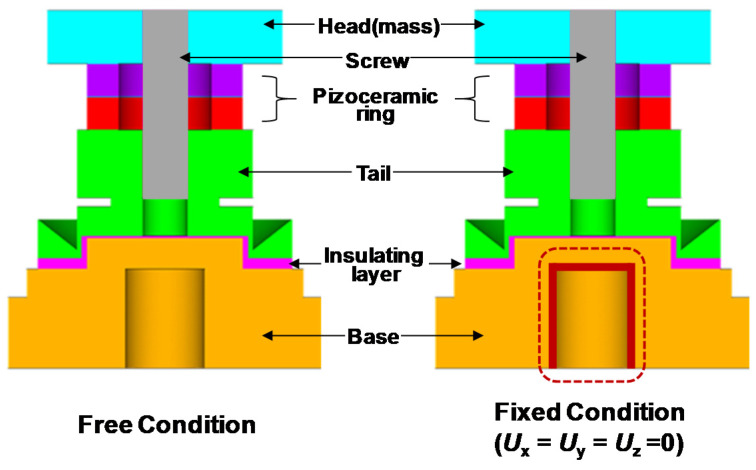
Basic structure of compression-type piezoelectric accelerometer and constituent components used for finite element modeling (**left**: Free condition, **right**: fixed condition).

**Figure 2 sensors-20-03545-f002:**
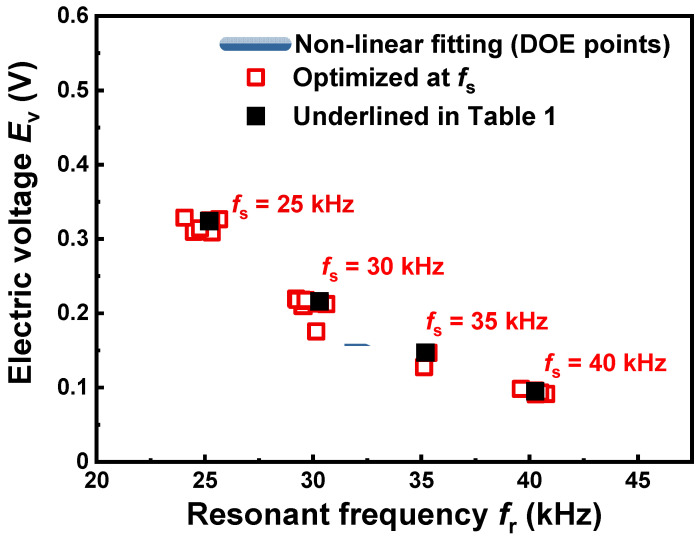
Optimization results of resonant frequency *f*_r_ and electric voltage *E*_v_ at different set values of resonant frequency *f*_s_, together with non-linear fitting curve of design of experiments (DOE) test points used for numerical optimization (from ref. [10]).

**Figure 3 sensors-20-03545-f003:**
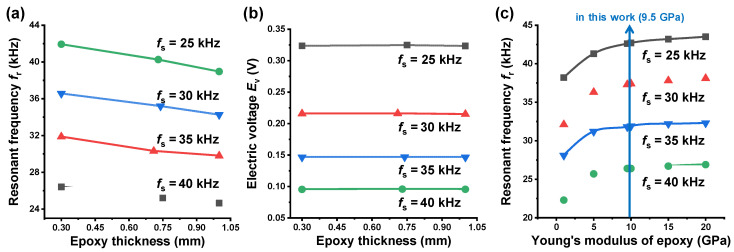
(**a**) Variation of resonant frequency *f*_r_ with epoxy thickness; (**b**) Variation of electric voltage *E*_v_ with epoxy thickness; (**c**) Dependence of Young’s modulus of epoxy on resonant frequency *f*_r_.

**Figure 4 sensors-20-03545-f004:**
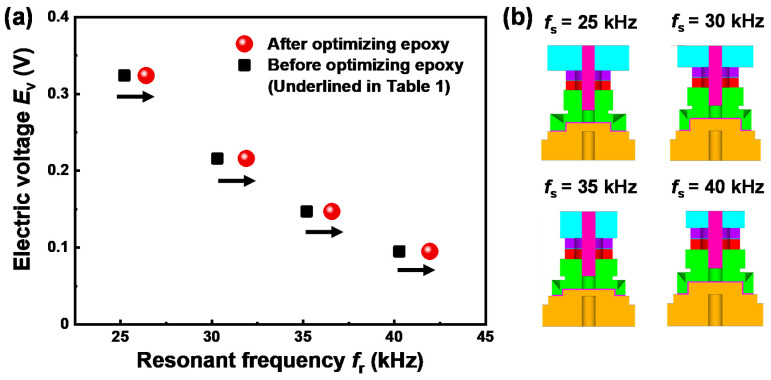
(**a**) Results of resonant frequency *f*_r_ and electric voltage *E*_v_ before (black squares) and after (red circles) optimizing epoxy thickness under conditions underlined in Table 1; (**b**) cross-sectional views for corresponding designs.

**Figure 5 sensors-20-03545-f005:**
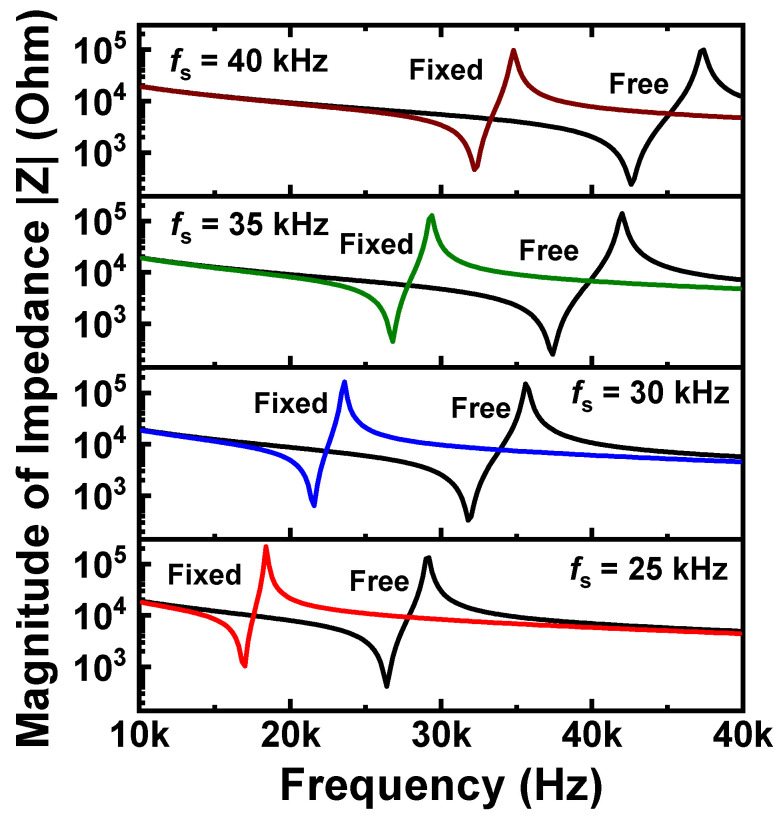
Impedance-frequency spectra obtained from numerical simulation using free and fixed boundary conditions for optimized designs (*f*_s_ = 25, 30, 35, 40 kHz).

**Figure 6 sensors-20-03545-f006:**
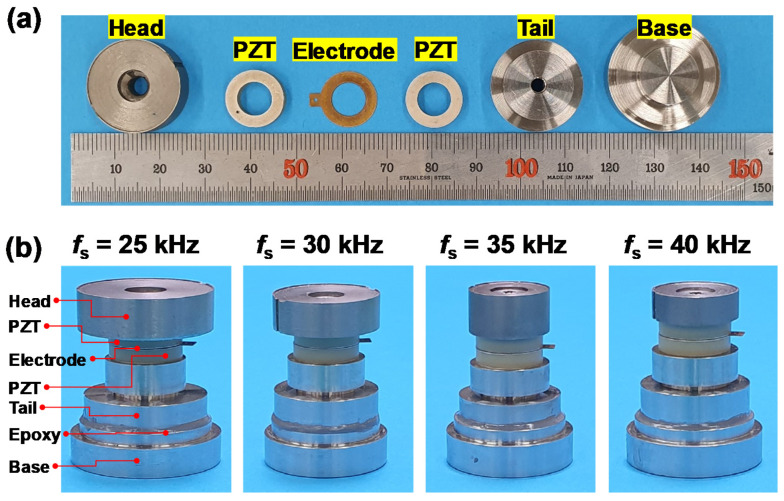
(**a**) Constituent components and PZT ceramic rings manufactured according to optimized design (*f*_s_ = 25 kHz); (**b**) Images of four accelerometer modules fabricated according to optimized designs (*f*_s_ = 25, 30, 35, 40 kHz).

**Figure 7 sensors-20-03545-f007:**
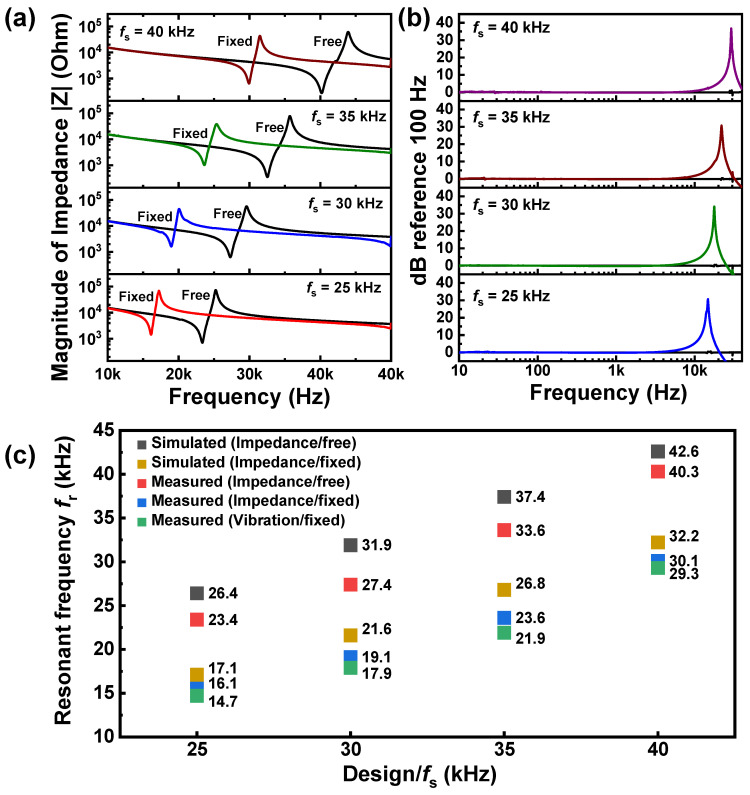
(**a**) Impedance versus frequency plots obtained from impedance measurements under free and fixed conditions; (**b**) Frequency response profiles obtained from vibration experiments under fixed condition; (**c**) Comparison between numerically simulated and experimentally measured resonant frequencies *f*_r_.

**Figure 8 sensors-20-03545-f008:**
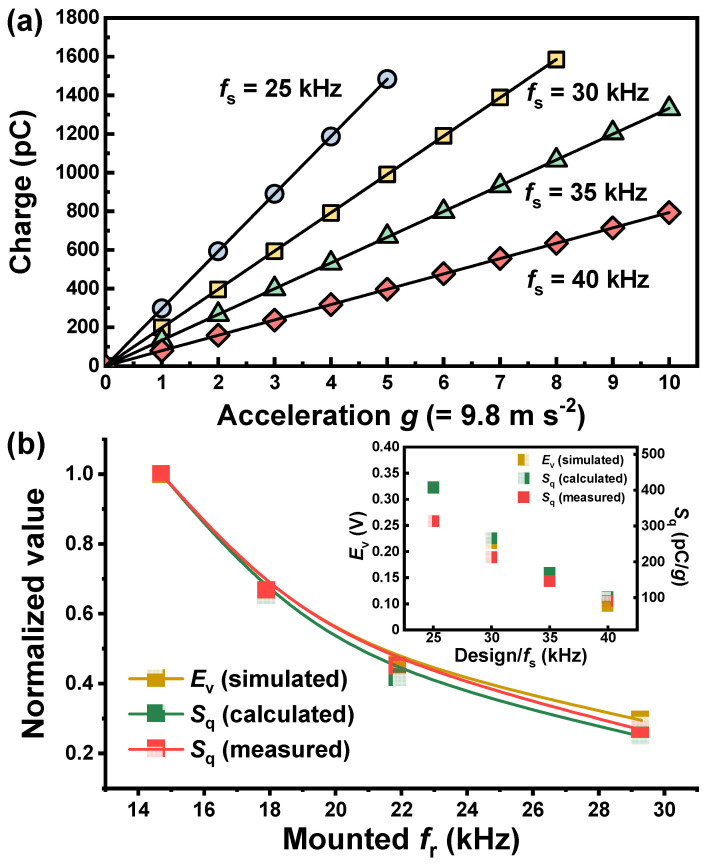
(**a**) Charge versus acceleration characteristics measured from accelerometer modules (test frequency = 159 Hz); (**b**) Variation of normalized values of *E*_v_, calculated and measured *S*_q_ with fixed *f*_r_ (inset shows design-dependent *E*_v_ and *S*_q_ values).

**Table 1 sensors-20-03545-t001:** Optimization results on design variables at different set values of resonant frequency *f*_s_ and corresponding results of resonant frequency *f*_r_ and electric voltage *E*_v_. Design variables are head OD (*x*_1_), head height (*x*_2_), piezoceramic OD (*x*_3_), piezoceramic ID (*x*_4_), piezoceramic thickness (*x*_5_), tail OD (*x*_6_), tail height (*x*_7_), base OD (*x*_8_), base height (*x*_9_), and epoxy thickness (*x*_10_).

*f*_s_ * (kHz)	Run no.	*x*_1_ (mm)	*x*_2_ (mm)	*x*_3_ (mm)	*x*_4_ (mm)	*x*_5_ (mm)	*x*_6_ (mm)	*x*_7_ (mm)	*x*_8_ (mm)	*x*_9_ (mm)	*x*_10_ (mm)	Results	
												*f*_r_ (kHz)	*E*_v_ (V)
	1	23.0	7.0	12.6	7.5	2.7	13.8	4.5	24.9	1.5	1.0	25.665	0.3265
	2	23.0	6.6	12.6	7.5	2.7	13.9	5.8	26.4	2.4	0.8	25.323	0.3089
	3	23.0	7.0	12.6	7.5	2.7	14.0	5.7	26.4	2.4	0.8	25.288	0.3246
25	4	23.0	6.6	12.6	7.5	2.7	13.2	5.8	26.4	2.4	0.8	24.501	0.3098
	5 **	23.0	7.0	12.6	7.5	2.7	13.9	5.7	26.4	2.4	0.8	25.203	0.3248
	6	23.0	7.0	12.6	7.5	2.7	13.2	4.5	26.4	3.5	0.8	24.063	0.3286
	7	23.0	6.7	12.6	7.5	2.7	13.5	5.8	26.4	2.4	0.8	24.781	0.3144
30	1	21.6	4.1	12.6	7.5	2.7	14.7	5.4	26.0	1.7	0.6	30.149	0.1755
	2	20.3	5.7	12.6	7.5	2.7	15.2	4.9	25.8	3.2	0.9	29.569	0.2099
	3 **	20.5	5.8	12.6	7.5	2.7	15.1	4.9	24.4	3.2	0.7	30.300	0.2161
	4	20.7	5.7	12.6	7.5	2.7	15.1	4.9	25.4	3.2	0.9	29.220	0.2196
	5	20.6	5.7	12.6	7.5	2.7	14.8	4.9	25.2	3.2	0.7	29.488	0.2180
	6	20.2	5.8	12.6	7.5	2.7	14.9	4.9	24.4	3.2	0.7	30.611	0.2122
	7	20.3	5.8	12.6	7.5	2.7	14.9	4.9	24.4	3.2	0.9	29.916	0.2131
	8	20.3	5.7	12.6	7.5	2.7	15.2	4.9	25.8	3.2	0.9	29.512	0.2097
	9	20.5	5.7	12.6	7.5	2.7	14.8	4.9	25.2	3.2	0.7	29.551	0.2172
	10	20.7	5.7	12.6	7.5	2.7	15.1	4.9	25.4	3.2	0.9	29.289	0.2180
	11	20.5	5.7	12.6	7.5	2.7	14.8	4.9	25.2	3.2	0.7	29.589	0.2168
	12	20.6	5.7	12.6	7.5	2.7	15.2	4.9	25.4	3.2	0.9	29.278	0.2180
	13	20.2	5.8	12.6	7.5	2.7	14.9	4.9	24.4	3.2	0.7	30.318	0.2147
	14	20.6	5.7	12.6	7.5	2.7	14.9	4.9	25.1	3.2	0.7	29.661	0.2179
	1	18.2	4.0	12.6	7.5	2.7	15.1	5.4	26.4	1.7	0.6	35.131	0.1276
35	2	15.1	7.0	12.6	7.5	2.7	15.2	4.5	24.4	3.5	0.6	35.334	0.1468
	3 **	15.1	7.0	12.6	7.5	2.7	15.1	4.5	24.4	1.6	0.7	35.191	0.1468
	1	15.0	4.0	12.6	7.5	2.7	15.1	5.4	26.4	1.7	0.6	40.297	0.0907
	2	15.0	4.0	12.6	7.5	2.7	14.7	4.7	24.4	3.0	0.7	40.770	0.0914
40	3	15.2	4.0	12.6	7.5	2.7	14.7	4.7	24.4	2.9	0.7	40.489	0.0934
	4 **	15.0	4.3	12.6	7.5	2.7	15.2	4.8	24.4	3.0	0.7	40.265	0.0958
	5	15.2	4.3	12.6	7.5	2.7	15.0	4.8	24.5	3.0	0.7	39.602	0.0982

* *f*_s_: Set value of resonant frequency used for numerical optimization of design variables. ** Underlined: Conditions chosen among the results optimized at different *f*_s_ values.

**Table 2 sensors-20-03545-t002:** Values of optimized design variables shown in Figure 4b and corresponding results on resonant frequency *f*_r_ and electric voltage *E*_v_ for free and fixed boundary conditions.

*f*_s_ (kHz)	*x*_1_ (mm)	*x*_2_ (mm)	*x*_3_ (mm)	*x*_4_ (mm)	*x*_5_ (mm)	*x*_6_ (mm)	*x*_7_ (mm)	*x*_8_ (mm)	*x*_9_ (mm)	*x*_10_ (mm)	Simulated Results			
											(Free-Mode)		(Fixed-Mode)	
											*f*_r_ (kHz)	*E*_v_ (V)	*f*_r_ (kHz)	*E*_v_ (V)
25	23.0	7.0	12.6	7.5	2.7	13.9	5.7	26.4	2.4	0.3	26.4	0.324	17.0	0.324
30	20.5	5.8	12.6	7.5	2.7	15.1	4.9	24.4	3.2	0.3	31.9	0.216	21.6	0.216
35	15.1	7.0	12.6	7.5	2.7	15.1	4.5	24.4	1.6	0.3	37.4	0.147	26.8	0.147
40	15.0	4.3	12.6	7.5	2.7	15.2	4.8	24.4	3.0	0.3	42.6	0.095	32.2	0.095

**Table 3 sensors-20-03545-t003:** Room-temperature physical properties of PZT ceramic rings.

Property at 25 °C	PZT Ceramic Ring
Weight *m* (g)	1.616
Density *ρ* (kg/m^3^)	7760
Outer diameter/O.D. (mm)	12.62
Inner diameter/I.D. (mm)	7.52
Thickness *t* (mm)	2.57
Remanent polarization *P*_r_ (μC /cm^2^)	15.9
Coercive field *E*_c_ (kV/cm)	14.7
Dielectric constant *ε*_r_	1.85 × 10^3^
Longitudinal piezoelectric coefficient *d*_33_ (pC/N)	400.0 ± 2.1
Electromechanical coupling factor *k*_p_	0.36
Curie temperature *T*_c_ (°C)	367.2

**Table 4 sensors-20-03545-t004:** Results of Pearson’s *r*, calculated and measured charge sensitivities *S*_q_, head weight *m*_s_ and fixed resonant frequency *f*_r_ for four optimized designs.

Design/*f*_s_ (kHz)	Acceleration Range *g* (=9.8 m s^−2^)	Head Weight *m*_s_ (g)	Charge Sensitivity *S*_q_ (pC/*g*)		Measured *f*_r_ */Fixed Mode (kHz)
			Calculated	Measured	
25	1–5	51.8	406.5	296.9	14.7
30	1–8	33.8	264.9	198.2	17.9
35	1–10	21.6	169.2	133.3	21.9
40	1–10	12.9	101.1	79.4	29.3

* The values are measured by vibration experiments.

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
