# Peer review of "A Theoretical and Empirical Investigation of Design Characteristics in a Pb(Zr,Ti)O3-Based Piezoelectric Accelerometer"

_sensors, 2020, doi:10.3390/s20123545_

Round 1
Reviewer 1 Report
Dear the Authors,
Your paper "A theoretical and empirial investigation of design characteristics in a PbZrTi03-based piezoelectric accelerometer" presents an experimental validation of previously developed numerical techniques by focusing on a few optimized designs. It is well written and organized with clear and easy to understand results. I recommend the paper for publication but do have a few comments for you to consider that may add to the discuss of the paper.
1) On line 119 you mention the results are sensitive to the epoxy layer. As it turns out, the epoxy has optimal properties, but did you consider other adhesives (or press-fitting) when designing the model? Could this help optimize the design?
2) In the paragraph starting on line 184 you mention the frequency is slightly lower for experimental results than numerical results and discuss a few possibilities including epoxy thickness. Is it possible to model your system with varying epoxy thickness in a single model to see how this affects the results or can you determine the expected epoxy thickness to reduce the frequency accordingly? If so, perhaps you can adjust the thickness in the experiment to counteract any effects.
3) Similarly to (2), could hysteresis in the PZT (or other components) cause a reduction in the frequency? At high frequencies this can induce large energy losses and increase the temperature of the system, changing the behavior of the PZT slightly (or drastically as Tc is approached). Could this result in the frequency reduction as well?
Again, I recommend the paper for publication but am curious if these comments may add to the discussion or introduce other possible design parameters to future studies.
Thank you,
The Reviewer
Reviewer 2 Report
The paper “A theoretical and empirical investigation of design characteristics in a Pb(Zr,Ti)O3-based piezoelectric accelerometer” is interesting, because in my opinion it can contribute to developing of the devices such as piezoelectric accelerometer. The paper shows us that models can be designed starting from the application.
Following the review of the paper “A theoretical and empirical investigation of design characteristics in a Pb(Zr,Ti)O3-based piezoelectric accelerometer”, we can conclude that:
- The paper does not contain the mathematical modeling part used for the numerical modeling process, respectively in the proposed numerical optimization. Although referencing reference [10], it is necessary to effectively add this section to the paper.
- Regarding the experimental results, you indicate in the paper, all type of the equipment being used in order to obtain the experimental values. But, it is necessary to clarify all the measurement conditions for achieving the experiments, for all experiments performed. It is necessary to present in the paper the whole scheme of measurements and experimentation in order to obtain all the experimental values. More aspects of the experimental method and also the working way are necessary to add in the paper, in order to obtain the frequency response characteristic and the charge sensitivity values. Additional figures in this respect would be desirable.
- Please explain in the paper for what reason you not considered for the piezoceramic ring the dispersion manufacturing which can influence the results, besides the difficulties in uniformly maintaining the epoxy thickness at the numerical size of 0.32 mm, because “the dimensions of piezoceramic ring were kept constant: 12.6 mm for the outer diameter (O.D.), 7.5 mm for the inner diameter (I.D.) and 2.7 mm for the thickness”. This is all the more so as it is used two piezoceramic rings that are connected electrically in parallel and mechanically in series.
- The electromechanical coupling factor kp, for piezoceramic material PZT that uses is 0.36, Table 3. Please explain in the paper how this parameter is taken into account in order to obtaining the electric voltage Ev, Figure 8 b. Please explain in the paper what are the consequences of the electromechanical coupling factor kp value for the piezoceramic material PZT that uses for obtaining a value of the electric voltage Ev?
- The paper contains a lot of value results that need to be highlighted in the conclusions. The conclusions of the paper should contain and the possible implications of this study in future practical developments. What are the prospects for capitalizing on this research? It is necessary to add these aspects to the conclusions.
Round 2
Reviewer 2 Report
The paper “A theoretical and empirical investigation of design characteristics in a Pb(Zr,Ti)O3-based piezoelectric accelerometer” in the revised form, is interesting for industry and science, because in my opinion it can contribute to developing of the devices such as piezoelectric accelerometer. The paper shows us that models can be designed starting from the application. Following the review of the paper “A theoretical and empirical investigation of design characteristics in a Pb(Zr,Ti)O3-based piezoelectric accelerometer”, in the revised form, it can be said that the paper has value through what it proposes, especially in the “Materials and Methods” section. A notable result as the authors themselves conclude, can be specified: “In addition, this study also suggests that the piezoelectric accelerometer can be selectively prepared using the optimized designs in accordance with the usage objective or the requirements with regard to the sensitivity and resonant frequency”.
Following the review of the paper “A theoretical and empirical investigation of design characteristics in a Pb(Zr,Ti)O3-based piezoelectric accelerometer”, in the revised form, I can conclude that:
I have looked carefully on the answers of the authors on all the recommendations, as well as the entire paper, in the revised form.
I think that the authors have correctly and thoroughly solved all the recommendations. Also, have been introduced within the paper the result paragraphs.
The paper is interesting and shows a high degree of novelty and future impact.
In my opinion, the manuscript has been significantly improved.
Congratulations to the authors for their proven scientific rigor and high professionalism.